# Assessment patient satisfaction towards emergency medical care and its determinants at Ayder comprehensive specialized hospital, Mekelle, Northern Ethiopia

**Goitom Molalign Takele** [1]*, **Negash Abreha Weldesenbet**[1], **Nahom Girmay**[2], **Habtamu Degefe**[2], **Rigbe Kinfe**[2]

1 Department of Emergency and Critical Care Nursing, School of Nursing, College of Health Sciences, Mekelle University, Mekelle, Ethiopia, 2 Ayder Comprehensive Specialized Hospital, College of Health Sciences, Mekelle University, Mekelle, Ethiopia

* goitommolalign49@gmail.com

## Abstract

### Background

As the healthcare industry shifts toward patient-centered models, providers will need to fully understand patient satisfaction and how they affect their practices. This study aimed to assess patient satisfaction towards the emergency medical care and factors associated with at Ayder specialized comprehensive hospital, Emergency room, Mekelle, Ethiopia.

### Methods

An institution-based cross-sectional study was conducted from March 1–30, 2019. A systematic random sampling method was used to enroll 299 study participants. Data were collected using a standard Brief Emergency Department Patient Satisfaction Scale questionnaire by trained data collectors. Data was entered into EpiData 3.1 then exported and analyzed by SPSS version 22. Binary and multiple logistic regression were used to assess the factors associated with patient satisfaction. Where the p-value of <0.05 was considered significant.

### Results

A total of 299 participants were enrolled in the study with a response rate of 99.3%. On overall patient satisfaction score majority (81.9%) of them were satisfied with the emergency medical care provided. The satisfaction rate towards emergency staff courtesy, emergency room environment, physician care satisfaction, general patient satisfaction, and patient family satisfaction was 80.3%, 37.5%, 75.9%, 70.9%, and 49.8% respectively. Those who arrived during the morning time of the day tend to be satisfied more with the emergency services (AOR = 4.8, 95% CI: 2.08, 11.4), while having low educational status (able to read and write) (AOR = 0.12, 95% CI: 0.03, 0.50) and waiting time till seen by a doctor (AOR = 1.3, 95% CI: 1.003, 1.4) was found to affect patient satisfaction negatively.

**Data Availability Statement:** All relevant data are within the paper and its Supporting information.

**Funding:** The author(s) received no specific funding for this work.

**Competing interests:** The authors have declared that no competing interests exist.

**Abbreviations:** ACSH, Ayder comprehensive specialized hospital; AOD, adjusted odds ratio; BEDSS, Brief emergency department satisfaction scale; CI, confidence interval; COR, crude odds ratio; ED, Emergency department; ER, emergency room; OPD, outpatient department; OR, odds ratio; SD, standard deviation; SPSS, statistical package for social science.

## Conclusions

The total patient satisfaction score towards emergency medical care was found to be good. The hospital management and emergency room staff should act on the identified factors especially on minimizing the patients waiting time to improve the quality of care in the emergency department.

## Introduction

Satisfaction refers to the good feeling that you have when you have achieved something or when something that you wanted to happen does happen [1]. The emergency room (ER) is the main gateway for patients visiting the hospital and with the highest turnover of patients where they came with acute, life-threatening medical needs. Satisfaction with emergency departments (ED) affects the overall satisfaction with hospitalization and the evaluation of patient satisfaction is a measurement of the quality of care in the ED [2, 3].

Although we couldn't keep all patients and their companions satisfied we can increase their satisfaction by working on the related indicators and trying to improve them [4, 5]. Among the factors to have affected patient satisfaction levels were: perceived waiting time, interpersonal skills/staff attitudes, provision of information/explanation, cleanliness of the emergency area, patient courtesy, nursing care, and patient-nurse communication [6–9].

Health care in developing countries has not traditionally focused on emergency medical care [10]. Emergency medicine is the newly developing department in the country facing different difficulties and with promising progress over the past years [11]. Since no previous studies were conducted in the ED of Ayder comprehensive specialized hospital (ACSH), assessing the patient level of satisfaction towards the emergency services provided and its determinants in will provide important input for identifying the gap and strengthening the potential area of opportunity [12].

## Material and methods

### Study setting

This study was conducted at ACSH, ED, Mekelle, Tigray, Ethiopia. ACSH is a tertiary care teaching university hospital, the main referral center in the Tigray region, northern Ethiopia with a catchment area of the whole Tigray region and neighboring regions of Amhara and Afar which gives all referral and non-referral services for more than 5 million peoples within the catchment area and has the busiest emergency department due to the lack of nearby tertiary care-giving hospitals. Every day, at least 200 patients visit the emergency department.

### Study design and study period

An institution-based, cross-sectional study design was conducted from March 1–30, 2019.

### Study subjects

Study participants were recruited using a systematic random sampling method from patients visiting the ED of ACSH during the data collection period. Those with the ability to communicate in common regional language (Tigrigna) were included in the study.

## Sample size calculation and sampling method

The sample size was determined using a single population proportion formula. A 95% confidence level, 5% margin of error and 77% satisfaction level of the patients were considered from a study conducted in Jimma [13]. By considering the non-response rate, we added 10% of the calculated sample a total of 301 sample size was assumed. A systematic sampling method was employed to recruit the study participants.

## Data collection tool and data collection procedure

Data were collected using a modified version of the structured Brief Emergency Department Patient Satisfaction Scale (BEDPS) which was developed to assess patient satisfaction towards the services provided in the ED [14]. The questionnaire which was 4 point Likert scale was modified into 5 point Likert scale to increase response options. Then the English version questionnaire was translated into the local language (Tigrigna) first then translated back into English. Pre-test was done in Mekelle general hospital on 15 patients with an internal consistency result of Cronbach's alpha 0.83. Overall the questionnaire contains: Socio-demographic part, medical characteristics of the patient and BEDPS 20 questions divided into five parts, 6 questions related to ED staff, 3 questions about ED environment, 4 questions about patient care satisfaction, 5 questions about general patient satisfaction, and 2 regarding patients' family satisfaction (S1 File). The total sample was distributed into different working shifts considering, busy work hours, different providers, day of the week and the type of client complaint were considered to affect satisfaction level. The total sample size was distributed to a different shift. To select participants in each shift, random numbers were used. The study subjects were interviewed face-to-face by three trained research assistants (graduating class Bsc. Nurse Students) immediately after getting an emergency service i.e. while they are being admitted to the inpatient ward, consulted to respective specialty units or before they were discharged to home after getting emergency medical services.

## Data quality control

To maintain the quality of the data, training was given to data collectors and the questionnaire was also pre-tested. Ten percent double data entry into Epi data was performed for data accuracy, and no difference was found on the outcome of double-entered data. During data collection, data collectors were not wearing a gown and were also not working in the ED to avoid the respondent bias. Collected data was checked for completeness every day by the research assistant team before data entry.

## Ethics approval and consent to participate

Ethical approval for the study was granted by Mekelle University, college of health sciences, school of Nursing, ethical committee ERC 1051/2019, on February 20, 2019. Then the letter of support from the ethical committee was submitted and accepted by the emergency department of the hospital. All study participants were notified about the purpose of the study, the right to refuse to participate in the study, and confidentiality of the information gathered. Written Consent was also obtained from each voluntary study participants and for age less than 18 years old consent was obtained from the parents or guardian.

## Statistical analysis

Data were first cleared and coded by the research assistant team then entered into Epidata version 3.1. and exported into SPSS version 22 for analysis. Descriptive statistics were used to determine the satisfaction level, calculate the socio-demographic characteristics and disease

profiles. Binary and multiple logistic regression were used to identify factors affecting the patient satisfaction level. Variables with p-value <0.05 on binary logistic regression analysis were subjected for multiple logistic regression analysis. Level of significance was declared at p-value <0.05. Result of the analysis were presented as Adjusted Odds Ratio (AOD) with its 95% confidence interval. In the multivariate logistic regression the Hosmer Lemeshow goodness of fit as test for model fitness was used.

### Operational definition

■ Patient satisfaction was measured by a Likert scale of 20 questions and was graded as Very dissatisfied (1), Dissatisfied (2), Fair/indifferent (3), Satisfied (4) and Very satisfied (5). Those scoring the mean or below were considered as dissatisfied while a score above the mean was labeled as satisfied.

## Results

### Socio-demographic characteristics of the respondents

From 301 participants approached 299 have completed the interview, making a response rate of 99.3% in which only two participant did not complete the interview due to unexplained reason. More than half 154 (51.2%) of them were males and the mean (SD) age of the study participants was 35 (15) years. For majority of the respondent 162 (54.2%) it was their first time visit to the hospital. Concerning their educational status being able to read and write 95 (31.8%) outnumbers the rest. The emergency department provided service in three shifts with more than half 164 (54.8%) of the study participants arrived at the ED during the night shift followed with evening and morning shifts, 74 (24.7%) and 61 (20.4%) respectively. Regarding the outpatient department (OPD) site visited in the ED, a majority 198 (66.2%) of them were checked in the medical emergency OPD. More than half (55.2%) of the interviewees were the patient themselves. With regard to the degree of confidence they feel to get good service in the future majority of them are confident and very confident, 113 (37.8%) and 131 (43.8%) respectively. One hundred twenty-seven (41.5%) of the patients were managed in the emergency department the rest 168 (56.1%) were admitted to the inpatient ward. The mean (SD) waiting time till seen by a doctor was 43 (67) minutes' ranging from 5 minutes to 3 hours [Table 1].

### Medical condition of the patients

Half of the study participants had a history of visits to the hospital before and 114 (38.1%) of the patients had stayed in the ED for 3–6 hours after getting emergency care [Table 2].

### Patient's satisfaction towards the ED services

The vast majority of respondents (81.9%) were satisfied with the emergency care services provided. In the five categories of services they got, the respondents satisfaction rate to the emergency staff courtesy, emergency department environment, physician care, general patient satisfaction and patients family satisfaction were 80.3%, 37.5%, 75.9%, 70.9% and 49.8% respectively [Table 3].

### Determinants of patient satisfaction towards emergency medical care

On binary logistic regression analysis variables like educational status, waiting time till seen by a doctor, visiting time of the day and were significantly associated with patient satisfaction. On the multiple logistic regression analysis educational status was found to be associated with the level of satisfaction. Where the odd of individuals who are able to read and write, reporting to

**Table 1. Socio-demographic characteristics and health services of the study participants at emergency OPD of ACHS (n = 299).**

| Variables | Groups | N (%) |
|---|---|---|
| **Sex** | Male | 154 (51.5) |
| | Female | 145 (48.5) |
| **Age (years)** | 1–10 | 33 (11) |
| | 11–20 | 30 (10) |
| | 21–30 | 116 (38.8) |
| | 31–40 | 52 (17.4) |
| | 41–50 | 27 (9.05) |
| | 51–64 | 27 (9.05) |
| | $\geq$ 65 | 14 (4.7) |
| **Educational status** | Illiterate | 66 (22.1) |
| | Able to read and write | 95 (31.8) |
| | Primary school | 18 (6) |
| | Secondary school | 45 (15) |
| | Diploma | 34 (11.4) |
| | Degree | 41 (13.7) |
| **Time of visit** | Morning | 61 (20.4) |
| | Evening | 74 (24.7) |
| | Night | 164 (54.9) |
| **Frequency of visit** | First time | 162 (54.2) |
| | Two and more | 137 (45.8) |
| **Respondent** | Patient him self | 165 (55.2) |
| | Attendant | 134 (44.8) |
| **OPD site visited** | Medical emergency | 198 (66.2) |
| | Surgical emergency | 64 (21.4) |
| | Pediatrics emergency | 37 (12.4) |
| **Subsequent management decision** | Managed in the ED | 127 (42.5) |
| | Admitted | 168 (56.1) |
| | Transferred | 4 (1.4) |
| **Degree of confidence to get good service in the future** | Very confident | 113 (37.8) |
| | Confident | 131 (43.8) |
| | Somewhat confident | 53 (17.7) |
| | Not confident | 2 (0.7) |
| **Do you feel discriminated** | Yes | 29 (9.7) |
| | No | 270 (90.3) |
| **Residency** | Urban | 197 (65.9) |
| | Rural | 102 (40.1) |
| **Waiting time till seen by Dr.(Mean ± SD in minutes)** | 43 ± 67 minutes | |

be satisfied were 88% less likely compared to the degree and above holders (AOR = 0.12, 95% CI: 0.03, 0.5, p = 0.002). Visiting time of the day had also an effect on the satisfaction of patients with emergency care provided (p < 0.0001). Where patients who arrived during the morning shift of the day, reporting to be satisfied were 4.8 times compared to their counterparts arriving at night shift (AOR = 4.8, 95% CI: 2.08, 11.4). Another vital determining factor of patient satisfaction was the waiting time till seen by the doctor. For every minute decrease in waiting time to be seen by a doctor, there is an increase in patient satisfaction by 1.3 times more (AOR = 1.3, 95% CI: 1.003, 1.4) [Table 4].

**Table 2. Medical condition characteristics of patients visiting the emergency department of ACSH (n = 299).**

| Medical characteristics | Category | N (%) |
|---|---|---|
| Presence of any past illness | Yes | 121 (40.5) |
| | No | 178(59.5) |
| Emergency visit before | Yes | 150(50.2) |
| | No | 149 (49.8) |
| Duration of stay in the ED after examination | 1–2 hours | 28 (9.4) |
| | 3–6 hours | 114 (38.1) |
| | 7–13 hours | 38 (12.7) |
| | 14–23 hours | 22 (7.4) |
| | ≥ 24 hours | 97 (32.4) |
| History of admission to hospital | Yes | 139 46.5) |
| | No | 160 (53.5) |
| Previous chronic illness | Yes | 153 (51.2) |
| | No | 146 (48.8) |
| | Hypertension | 14 (4.7) |
| | Cardiac problem | 22 (7.4) |
| | Cancer | 17 (5.7) |
| | Diabetes mellitus | 24 (8) |
| | Asthma | 24 (8) |
| | Others | 52 (17.4) |

## Discussion

In the present study the overall patient satisfaction towards the emergency medical service provided in ACSH was assessed using 20 questions containing brief emergency department patient satisfaction scale. Where 81.9% of respondent have reported to be satisfied and factors like educational status of being able to read and write, arrival time at the morning shift of the day and waiting time till seen by a doctor were the determinants of patient satisfaction.

Majority (54.8%) of the patients had arrived in the emergency department during the night time of the day, which is similar to the study conducted in Gondar [6], whereas the data from study conducted in Hawassa showed that the majority of the cases had arrived during the morning shift of the day [15]. The mean waiting time till seen by a physician in the ED was 43 minutes, ranged 5 minutes—6 hours, which is in line with a study conducted in central Saudi Arabia, Morocco and Imam Raza hospital [16–18], but found to be higher when compared to the study conducted in Iran 10.7± 6.1 minutes [19]. The lengthy waiting time in the current study might be due to the burden of the high patient flow to the hospital since this is the only referral hospital with a wide catchment area.

**Table 3. The overall satisfaction rate of study participant towards the emergency medical services in ACSH (n = 299).**

| Questions | Satisfied (%) | Dissatisfied (%) |
|---|---|---|
| The overall satisfaction of ED staff courtesy | 240(80.3) | 59(19.7) |
| The overall satisfaction of ED environment | 112(37.5) | 187(62.5) |
| The overall satisfaction of physician care | 227(75.9) | 72(24.1) |
| The overall score of general patient satisfaction | 212(70.9) | 87(29.1) |
| The overall satisfaction patient's family care | 149(49.8) | 150(50.2) |
| **Total overall satisfaction level of the patients towards the ED services** | **245(81.9)** | **54(18.1)** |

**Table 4. Logistic regression analysis of factors associated with patient satisfaction at the emergency OPD in ACSH (n = 299).**

| Variables | Level of satisfaction | | OR (95% CI) | | P–value |
|---|---|---|---|---|---|
| | Satisfied | Not satisfied | Crude | Adjusted | |
| **Educational status** | | | | | 0.001* |
| Illiterate | 66 | 22.1 | 1.6(0.6, 4.1) | 0.7 (0.2,1.9) | 0.741 |
| Able to read & write | 95 | 31.8 | 0.1(0.03, 0.4) | 0.12 (0.03,0.5) | 0.002* |
| Primary school | 18 | 6 | 0.7(0.2,2.8) | 0.8 (0.2,3.5) | 0.875 |
| Secondary school | 45 | 15 | 1.6(0.6, 4.1) | 2.0 (0.7,5.8) | 0.131 |
| Diploma | 34 | 11.4 | 0.6(0.1, 1.7) | 0.8 (0.2,2.8) | 0.999 |
| Degree & above | 41 | 13.7 | 1 | 1 | |
| **Time of visit** | | | | | 0.001* |
| Morning | 61 | 20.4 | 5.2 (2.5, 10.8) | 4.8 (2.08, 11.4) | <0.0001* |
| Evening | 74 | 24.7 | 2.5 (1.19, 5.4) | 2.2 (0.9, 5.2) | 0.061 |
| Night | 164 | 54.8 | 1 | 1 | |
| **Waiting time till seen by doctor(mean± SD)** | 43 ± 67 minutes | | 1(1.004,1.035) | 1.3(1.003,1.4) | 0.002* |

*p- value < 0.05, OR: odds ratio, COR: crude odds ratio, AOD: adjusted odds ratio, SD: standard deviation, CI: confidence interval

The overall satisfaction score of the patients with the services rendered at ACSH, ED majority (81.9%) of them were satisfied, which is similar to the studies conducted in southern Ethiopia, Hawassa 86.7%, and Jimma 78% [13, 15]. Whereas, higher compared to the study conducted in Gondar, 51.7%, [6] and Morocco 66% [16]. This difference in level of satisfaction might be due to the difference in the tool used to measure patient satisfaction, and difference in clinical characteristics (medical condition on arrival) of the patients.

With regards to the five domains of patient satisfaction measurement, the respondents have lower satisfaction in the emergency department environment and patients' family satisfaction 37.5% and 49.8% respectively. Similarly, a study from Iran had also showed a lowest satisfaction towards pleasantness of the waiting area [19]. This shows a potential area of weakness and the hospital management should work in this area to improve the quality of care. As it has been stated in other studies providing a comforting environment, interpersonal skills in terms of courtesy, respect by health care providers in addition to communication skills, explanation, and clear information, which are more essential and influential than other technical skills such as clinical competency and hospital equipment [19, 20].

The time of the day where the patient arrives in the ED had an association with patient satisfaction. Where patients who arrived during the morning shift of the day were 4.8 times more likely to be satisfied compared to their counterparts arriving at night. In line with this a study by Zohrevandi [21], and press Graney report [22] found that patients who arrived during the morning shift of the day were more likely to be satisfied compared to their counterparts arriving at the night shift. This might be due to the additional number of staffs (senior physicians, medical and nursing students) during the morning shift of the day may give an extra advantage to the regular working staffs since this is a teaching hospital. While during the night shift those extra staff members may not be assigned to the night shift, and as the majority of the patient (54.8%) had arrived to the ED at nightshift overcrowding of the ED might have also its own effect.

This study revealed that a patient's waiting time till seen by a doctor was found to be negatively affecting patient satisfaction. For every minute decrease in patients waiting time to see the doctor, the satisfaction level will be increased by 1.3 times. This is similar to other studies and working on minimizing the waiting time could enhance patient satisfaction at the ED [16, 18, 23, 24].

Another determining factor associated with patient satisfaction was educational status were those patients with lower educational status (able to read and write) were less likely to be satisfied than their counterparts of degree and above holders which is in line to the study conducted in Morocco which was justified as patients with higher educational status might have a better understanding, able to settle medical debates easily and had accepted their care was dependent on good management [16].

## Strength and limitations

As a strength of this study, it is the first study to be conducted at the emergency department in the Tigray region, northern Ethiopia, and this would have a great input for clinicians and quality improvement of the hospital in terms of identifying potential areas of improvements and plan strategies of the care provided at the ED. Another strength of this study is the use of logistic regression will help to avoid confounding effect of the variables. With regard to the limitations we believe that there might be some confounding variables that we did not include in the data collection tool that could have an effect on patient satisfaction. First, Patients with different presentations might have different satisfaction rates, and the severity level of patients' case may influence satisfaction rates, e.g., people who are in a great deal of pain are likely to be dissatisfied. Second, the responses by the companion and the patient him/herself were not assessed separately which might have some difference. In case of comatose/patients in distress or pediatric patients, relatives or patient attendants were asked to fill the questionnaire might have its own limitation in expressing the true feeling of the patient opinion. Third, the study was conducted in one site; thus, the results cannot be generalized to all Tigray region Hospitals. Fourth, patients were not stratified with regards to whether surgical interventions were performed or not at the ED i.e. patient satisfaction might have been influenced by surgical interventions done to some in need against those medically treated.

## Conclusion

In summary, the majority of patients (81.9%) were satisfied with the overall emergency medical care received at the ED. The lowest satisfaction level was recorded in the emergency department environment and patient family satisfaction measures domains. Determinant factors of patient satisfaction identified were: educational status, waiting time till seen by the doctor and the time of the day visited (morning). The hospital management and staff at the ED should work together to minimize the waiting time.

## Supporting information

**S1 Data.**
(SAV)

**S1 File.**
(DOCX)

## Acknowledgments

We would like to acknowledge study participants and Emergency department staffs for their cooperation during data collection.

## Author Contributions

**Conceptualization:** Goitom Molalign Takele, Negash Abreha Weldesenbet, Rigbe Kinfe.

**Data curation:** Habtamu Degefe, Rigbe Kinfe.

**Formal analysis:** Goitom Molalign Takele, Nahom Girmay, Habtamu Degefe.

**Investigation:** Goitom Molalign Takele, Negash Abreha Weldesenbet.

**Methodology:** Goitom Molalign Takele, Negash Abreha Weldesenbet.

**Software:** Goitom Molalign Takele.

**Writing – original draft:** Nahom Girmay, Habtamu Degefe.

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
