## [Decision Letter · Decision Letter 0]

21 Aug 2020

PONE-D-20-06847

Assessment Patient satisfaction towards emergency medical care and its determinants at Ayder comprehensive specialized hospital, Mekelle, Northern Ethiopia

PLOS ONE

Dear Dr. Molalign,

Thank you for submitting your manuscript to PLOS ONE; I sincerely apologise for the unusually delayed review timeframe. After careful consideration, we feel that it has merit but does not fully meet PLOS ONE’s publication criteria as it currently stands. Therefore, we invite you to submit a revised version of the manuscript that addresses the points raised during the review process.

Your study has been assessed by four reviewers, whose comments are appended below. Although the reviewers to find the study to be of interest, they raise several points that must be addressed before further consideration of this work. Among the concerns are whether the questionnaire was pretested or validated, whether the study received approval from a dedicated research ethics committee or institutional review board, a discussion of the strengths and limitations of this work, and a comparison of these findings to similar studies in other regions. In addition, the reviewers raise several concerns about the statistical analyses that must be addressed.

We look forward to receiving your revised manuscript.

Kind regards,

Emily Chenette

Deputy Editor-in-Chief

PLOS ONE

Journal Requirements:

2.Please explain how the BEDPSS questionnaire was adopted and modified for this work. In addition, please state how the questionnaire was pre-tested and validated. If this did not occur, please include the rationale for not pre-testing or validating the questionnaire.

We note that ethics approval was granted by Mekelle University, College of Health Sciences, School of Nursing. It is not clear if this group is duly qualified to provide ethics approval for human subjects research. Could you please clarify whether the study was approved by an Institutional Review Board or Research Ethics Committee, include the date on which the study was approved and provide the approval or permit number that was issued?

3.We suggest you thoroughly copyedit your manuscript for language usage, spelling, and grammar. If you do not know anyone who can help you do this, you may wish to consider employing a professional scientific editing service.  

4.We note that Table 3, Table 5 and Additional File 1  in your submission (which seem to reproduce the "Brief Emergency Department Patient Satisfaction Scale questionnaire" (http://www.journals.sbmu.ac.ir/AAEM/index.php/AAEM/article/view/278/580) contain copyrighted information. All PLOS content is published under the Creative Commons Attribution License (CC BY 4.0), which means that the manuscript, images, and Supporting Information files will be freely available online, and any third party is permitted to access, download, copy, distribute, and use these materials in any way, even commercially, with proper attribution. For more information, see our copyright guidelines: http://journals.plos.org/plosone/s/licenses-and-copyright.

1.         You may seek permission from the original copyright holder of Table 3, Table 5 and Additional File 1 to publish the content specifically under the CC BY 4.0 license.

5.We note that you have indicated that data from this study are available upon request. PLOS only allows data to be available upon request if there are legal or ethical restrictions on sharing data publicly. For information on unacceptable data access restrictions, please see http://journals.plos.org/plosone/s/data-availability#loc-unacceptable-data-access-restrictions.

6. Your ethics statement must appear in the Methods section of your manuscript. If your ethics statement is written in any section besides the Methods, please move it to the Methods section and delete it from any other section. Please also ensure that your ethics statement is included in your manuscript, as the ethics section of your online submission will not be published alongside your manuscript.

<h1>** **</h1>

Reviewers' comments:

Reviewer's Responses to Questions

**Comments to the Author**

1. Is the manuscript technically sound, and do the data support the conclusions?

Reviewer #1: Yes

Reviewer #2: Yes

Reviewer #3: Partly

Reviewer #4: Partly

2. Has the statistical analysis been performed appropriately and rigorously? 

Reviewer #1: Yes

Reviewer #2: Yes

Reviewer #3: N/A

Reviewer #4: No

3. Have the authors made all data underlying the findings in their manuscript fully available?

Reviewer #1: Yes

Reviewer #2: Yes

Reviewer #3: Yes

Reviewer #4: Yes

4. Is the manuscript presented in an intelligible fashion and written in standard English?

Reviewer #1: Yes

Reviewer #2: Yes

Reviewer #3: No

Reviewer #4: No

5. Review Comments to the Author

Reviewer #1: manuscript technically sound was good, and data supported the conclusions. Additionally the statistical analysis been performed appropriately. The Tool need only to be clarified , the validity and reliability if did it by author or it was standardize and valid by the established author and when.

Reviewer #2: I have read this paper with great interest. The study assessed the degree of patient satisfaction regarding the Emergency Medical Services. The authors found the total patient satisfaction score towards emergency medical care to be good, as they also determined that the patient's waiting time is considered to be significantly associated with patient satisfaction level. Other research studies also identified waiting time in the clinic as an important indicator of patient satisfaction. I did not identify any of major or minor concern. In conclusion, the research study has identified some important elements that are imperative to form the model of Emergency Medical Services.

Reviewer #3: The study seems interesting in the context of northern Ethiopia and it can provide insights to the policy makers who are involved in quality of care.

My comments to the manuscript are as such.

1. Most of the sentences are in passive voice; I would better prefer to use active voice especially in method section. like:

"Every day, 200 patients visit the emergency department of the hospital" instead of "Daily it is visited by up to 200 patients."

2. Why this sentence is in sample size calculation and sampling method.? I am quite confused about it. "Busy work hours, different providers, day of the week and the type of client complaint were considered to affect satisfaction level."

3. Do avoid using didn't : instead use did not ( finding section)

4. Do use past tense consistently in the finding section; you are vacillating from past tense to present tense.

5. I think interpretation of odds ratio is not appropriate: It should be like this: "The odds of patient who arrived during the morning time of the day, reporting to be satisfied were 4.8 times compared to their counterparts arriving at night ( AOR=4.8, 95% CI: 2.08, 11.4)." Do revisit interpretation of odds ratio minutely.

6. You calculated Adjusted Odds ratio, but you did not mention what are the independent variables you adjusted for? Do mention clearly about it.

7. I think it would be better to summarize your main finding in the first paragraph of the discussion section and then start to compare your findings with other studies.

8. What might be the reason of variation in mean waiting time between your study and Iran's study?

9. "This difference might be due to the difference in the tool used to measure patient satisfaction, difference in clinical characteristics (medical condition on arrival) of the patients and other different factors." In this sentence, other different factors means what, like?

10. It would be better to discuss why patient visiting in morning time were more satisfied comparing with other studies.

11. Why the study finding is contrary to the finding of study in Morroco in terms of educational status, as a predictor of patient satisfaction?

Reviewer #4: The Authors propose the hospital-based study to assess patient satisfaction towards emergency medical care and its determinants in Northern Ethiopia. The study designs and methods used are basically appropriate, and the interpretations of the results are reasonable. However, there are several areas from statistical and practical viewpoints where the manuscript needs to be strengthened.

1. Please indicate the response rate of this study.

2.A statement detailing including the reference number where appropriate of ethics committee, should appear in the manuscript.

3.How are the reliability and validity of the questionnaire.

4.Tables should also appropriately labeled to show the baseline results between respondents and non-respondents.

5. How the variables are selected in the logistic regression in Table 4? Please also show the goodness-of-fit results.

6.Please consider the comparison with the other epidemiological studies in other areas using table so make clear the significance of this study.

7.What is the originality and strengths of this study? How physicians or policy makers can deliberate with subjects based on the key findings of this paper?

8. Please make sure whether formats are described of references according to the instructions for authors.

Totally, I would like to congratulate the authors for the enthusiasm invested in this study. However, the manuscript does not reach the level of quality required for publication as original article without major revision in PLOS ONE.

6. PLOS authors have the option to publish the peer review history of their article (what does this mean?). If published, this will include your full peer review and any attached files.

Reviewer #1: **Yes: **Dr. Noha Mohamed Ibrahim Rashed

Reviewer #2: **Yes: **Tatjana Kitić Jaklič

Reviewer #3: No

Reviewer #4: **Yes: **Tao-Hsin Tung

---

## [Author Response · Author response to Decision Letter 0]

16 Oct 2020

Response to reviewers

In general the comments and recommendations were accommodated as to the journal protocol and submission guidelines. Supplementary information were also attached and the concern raised with regards to the data availability, ethical committee concern, and copy right policy to replicate the tables used Table 4 (modified) and Table 5 (removed) were addressed. Our response to respective reviewers are stated below.

Response to reviewer one: 

The tool has been pre-tested in Mekelle hospital Emergency department two weeks before the actual data collection and then tested for the internal reliability of the tool with Cronbach’s alpha result of 0.83.

Response to reviewer two: No comment or question has been raised.

Response to reviewer Three: 

all your comments and recommendations are accepted and we have made an adjustment to the final manuscript. Table below here is the response to the questions raised. 

Questions and Responses

1. Most of the sentences are in passive voice; I would better prefer to use active voice especially in method section. like:"Every day, 200 patients visit the emergency department of the hospital" instead of "Daily it is visited by up to 200 patients." = We have changed some of the sentences that were written in passive voice. 

2. Why this sentence is in sample size calculation and sampling method? I am quite confused about it. "Busy work hours, different providers, day of the week and the type of client complaint were considered to affect satisfaction level."= The sentence has been moved from the sample size calculation section to the data collection procedure section. We considered it to affect patient satisfaction because in study conducted in Gondar they found the satisfaction to be affected by the day of the week visited, and busy work hour which was morning shift and another study also found that the type of complaint they present with could affect their satisfaction. So that considering these recommendations we tried to allocate the sample in each shifts, to get as diverse response as possible. 

3. Do avoid using didn't : instead use did not ( finding section)= Have been corrected 

4. I think interpretation of odds ratio is not appropriate: It should be like this: "The odds of patient who arrived during the morning time of the day, reporting to be satisfied were 4.8 times compared to their counterparts arriving at night ( AOR=4.8, 95% CI: 2.08, 11.4)." Do revisit interpretation of odds ratio minutely.=The interpretations of the odds ratio were also corrected as to your recommendation. 

5. You calculated Adjusted Odds ratio, but you did not mention what are the independent variables you adjusted for? Do mention clearly about it.= We added a sentence describing about the independent variables adjusted for in the result at the determinant of patient satisfaction section.

6. I think it would be better to summarize your main finding in the first paragraph of the discussion section and then start to compare your findings with other studies. = The result were summarized in the first paragraph of the discussion.

7. What might be the reason of variation in mean waiting time between your study and Iran's study?=The lengthy waiting time in the current study might be due to the burden of the high patient flow to the hospital since this is the only referral hospital with a wide catchment area.

9. "This difference might be due to the difference in the tool used to measure patient satisfaction, difference in clinical characteristics (medical condition on arrival) of the patients and other different factors." = In this sentence, other different factors means what, like? It was a typing error we took it out and only difference in the tool used to measure patient satisfaction, and difference in clinical characteristics (medical condition on arrival) of the patients were stated as possible reasons. 

10. It would be better to discuss why patient visiting in morning time were more satisfied comparing with other studies. =We have included findings from other studies who also claimed that the satisfaction rate of patients was higher for those who visited the ED at the morning shift compared to the night shift. 

11. Why the study finding is contrary to the finding of study in Morroco in terms of educational status, as a predictor of patient

Satisfaction?=It was written as “contrary” which was not correct as its odds ratio showed the result was similar with that of Morroco and the reason have been justified. 

Response to reviewer four:

All your comments and recommendations are accepted and we have made an adjustment to the final manuscript. Table below here is the response to the questions raised. 

Questions Responses 

1. Please indicate the response rate of this study. =Response rate were stated in the result section of the manuscript which is 99.3%.

2. A statement detailing including the reference number where appropriate of ethics committee, should appear in the manuscript.=The reference number for ethical review was provided in the manuscript. 

3. How are the reliability and validity of the questionnaire?=The questionnaire was translated both forward and backward then The internal consistency reliability was estimated with Cronbach’s α result of 0.83.

4. Tables should also appropriately labeled to show the baseline results between respondents and non-respondents.=The tables were adjusted to state the non-response rate as (n=299)

5. How the variables are selected in the logistic regression in Table 4? Please also show the goodness-of-fit results.=The variables on binary logistic regression with p-value <0.05 were subjected into multiple logistic regression. The Hosmer- Lemeshow goodness-of-fit result of (=0.737) was also presented at the Table 4. legend

6. Please consider the comparison with the other epidemiological studies in other areas using table so make clear the significance of this study.=We tried to compare with study in Gondar northwest of Ethiopia, Hawasa in the southern of Ethiopia and Jimma in southwest Ethiopia.

7. What is the originality and strengths of this study? How physicians or policy makers can deliberate with subjects based on the key findings of this paper?=Strength: this is the first study conducted in the Tigray region northern Ethiopia and would have a great input for clinicians and quality improvement of the hospital in terms of identifying potential areas of improvements and to plan strategies of the care provided at the ED accordingly.

---

## [Editor Report · Decision Letter 1]

26 Nov 2020

Assessment Patient satisfaction towards emergency medical care and its determinants at Ayder comprehensive specialized hospital, Mekelle, Northern Ethiopia

PONE-D-20-06847R1

Dear Dr. molalign,

We’re pleased to inform you that your manuscript has been judged scientifically suitable for publication and will be formally accepted for publication once it meets all outstanding technical requirements.

Kind regards,

Tao-Hsin Tung, PhD

Guest Editor

PLOS ONE

Additional Editor Comments (optional):

I am pleased to accept the revised version now.
---

## [Editor Report · Acceptance letter]

28 Dec 2020

PONE-D-20-06847R1 

Assessment Patient satisfaction towards emergency medical care and its determinants at Ayder comprehensive specialized hospital, Mekelle, Northern Ethiopia 

Dear Dr. Molalign Takele:

I'm pleased to inform you that your manuscript has been deemed suitable for publication in PLOS ONE. Congratulations! Your manuscript is now with our production department. 

Kind regards, 

on behalf of

Dr Tao-Hsin Tung 

Guest Editor

PLOS ONE